# Cellular Localization of Selected Porphyrins and Their Effect on the In Vitro Motility of Human Colon Tumors and Normal Cells

**DOI:** 10.3390/molecules28072907

**Published:** 2023-03-23

**Authors:** Maciej P. Frant, Mariusz Trytek, Kamil Deryło, Mateusz Kutyła, Roman Paduch

**Affiliations:** 1Department of Swine Diseases, National Veterinary Research Institute, Al. Partyzantów 57, 24-100 Puławy, Poland; 2Department of Industrial and Environmental Microbiology, Institute of Biological Sciences, Maria Curie-Skłodowska University, 20-033 Lublin, Poland; 3Department of Molecular Biology, Institute of Biological Sciences, Maria Curie-Skłodowska University, 20-033 Lublin, Poland; 4Department of Virology and Immunology, Institute of Biological Sciences, Maria Curie-Skłodowska University, Akademicka 19, 20-033 Lublin, Poland; 5Department of General and Pediatric Ophthalmology, Medical University of Lublin, Chmielna 1, 20-079 Lublin, Poland

**Keywords:** porphyrin, human colon tumor and normal cell lines, PDT, cell migration, cellular localization

## Abstract

Standard therapies for colorectal cancer cannot eliminate or sufficiently reduce the metastasis process. Photodynamic therapy (PDT) may be an alternative to minimizing this problem. Here, we examined the cellular localization of selected porphyrins and determined whether free-base and manganese (III) metallated porphyrins may limit colon cancer cells’ (HT29) or normal colon epithelial cells’ (CCD 841 CoTr) motility in vitro. White light irradiation was used to initiate the photodynamic effect. Porphyrin uptake by the cells was determined by porphyrin fluorescence measurements through the use of confocal microscopy. Free-base porphyrin was found in cells, where it initially localized at the edge of the cytoplasm and later in the perinuclear area. The concentrations of porphyrins had no effect on cancer cell migration but had a significant effect on normal cell motility. Due to the low concentrations of porphyrins used, no changes in F-actin filaments of the cellular cytoskeleton were detected. Signal transmission via connexons between neighbouring cells was limited to a maximum of 40 µm for HT29 and 30 µm for CCD 841 CoTr cells. The tested porphyrins differed in their activity against the tumor and normal cells’ migration capacity. Depending on the porphyrin used and the type of cells, their migration changed in relation to the control sample. The use of white light may change the activity of the porphyrins relative to the migratory capacity of the cells. The aim of the present study was to analyse the intracellular localization of tested porphyrins and their influence on the mobility of cells after irradiation with harmless white light.

## 1. Introduction

Colorectal cancer is one of the most common types of cancers. In the process of its development, an important role is played by the metabolic reprogramming of cells, leading to a gradual growth in tissue. Genetic changes, as well as phenomena occurring in the microenvironment of a growing tumor, significantly affect the speed of its growth, as well as its progress through subsequent stages of its development up to metastasis [1,2]. The presence of metastases significantly shortens median survival time (MST), despite the use of aggressive therapeutic methods. Tumor metastasis is therefore a significant challenge in both biological and clinical research [3]. One of the therapeutic methods is limiting the mobility of tumor cells and, thus, reducing the possibility of cell migration, both in the tissue and to distant organs [4]. However, classical therapeutic methods are not sufficient enough to reduce or eliminate the epithelial-mesenchymal transition (EMT) phenomenon and cell mobility [5]. There have been a lot of different types of chemical compounds used in recent studies, as potential anticancer drugs, such as, spirothiazolidines analogs [6], metal–organic frameworks (MOFs) [7], imidazole-triazole-glycoside hybrids [8] and porphyrins [9,10].

In this paper, we attempted to highlight porphyrins as promising alternatives for limiting cell movement and delaying the phenomenon of metastasis. Porphyrins are good candidates for photosensitizers in photodynamic therapy [9]. PDT is a promising therapeutic approach due to its ability to cause the rapid killing of cancer cells, as well as stimulate an anti-tumor immune response [10]. The activation of photosensitizers by the appropriate wavelength of visible light induces the production of free oxygen radicals, changes the tumor microenvironment, and finally, kills tumor cells [11]. PDT therapy also shows high selectivity with minimal invasiveness. This has made it possible to apply it to a substantially personalized approach to treating patients with colorectal cancer. In addition to the already indicated direct cytotoxic effects, PDT therapy may also limit the spread of neoplastic cells through indirect vascular effects [12,13]. Among cancers which are the most common in humans, the third is colorectal cancer. In 2020, there were about 0.9 million deaths from this cancer worldwide (with 1.9 million cases). This estimation indicates that the number of cases of colorectal cancer might even double by 2040 [14]. Despite the fact that colorectal cancer is located inside the human body, which makes it difficult for the delivery of light, there have been trials with porphyrin-based substances, which have given promising results, as photosensitizers in PDT [15,16,17]. Although the most efficient porphyrin-based photosensitizers lack coordinated metal ions, several metalloporphyrins (including, Zn, Pd, Sn, Ru, Gd and Al) have, however, been developed for clinical purposes. This is because of their properties, e.g., improved solubility and stability, which make them interesting therapeutic substances [18]. Chelation of a metal by a porphyrin ring can also potentially increase the generation of reactive oxygen species [19]. Manganese porphyrins have been widely investigated in many fields, including chemical catalysis, materials science and medical treatments. Proteomic analysis indicated that [Mn(TPP)Cl] impaired/disrupted several physiological processes, including DNA synthesis, transcription, mitochondrial respiration, RNA translation and immune response [20]. Manganese (III) meso-tetraphenylporphyrin chloride ([Mn(TPP)Cl]) has also been found to increase anion permeability in lung epithelial cells [20]. In turn, 5,10,15,20-Tetrakis (1-methyl-4-pyridinio) porphyrin is reported to be a promising photodynamic therapy photosensitizer. Moreover, it is one of the most studied G-quadruplex porphyrin ligands, used as a model to show that ligands can exhibit different binding features with different conformations of a human telomeric specific sequence [21]. However, the effects of both porphyrins have not been investigated on cell migration in a human colon tumor and normal cell cultures in the context of the effect of white light irradiation. The aim of our research was to assess the potential extent of limiting the mobility of human colorectal cancer cells and normal human intestinal epithelial cells under the influence of selected porphyrins (free-base and metallated porphyrins) (free-base and metallated ones) irradiated with harmless white light. We also determined the penetration and localization of the porphyrins into colon cells. We decided to test porphyrins with a similar structure (Mn porphyrin and FB porphyrin) to compare the different effects and factors involved in cancer and normal cell death/apoptosis under the influence of metallated and free-base porphyrins. Additionally, we wanted to observe whether some changes do not result from the intrinsic nature of porphyrins and not only from phototoxic effects, and different photochemical reactions. The conducted tests may be helpful in the diagnosis of colorectal cancer, as well as an alternative to limiting the spread of this cancer as early as at the screening stage.

## 2. Results

### 2.1. Cellular Localization of Porphyrins

Time-lapse, live-cell imaging experiments, using a laser scanning confocal microscope, were carried out to investigate the subcellular localization of the studied porphyrins in both cell lines. Endogenous porphyrins (autofluorescence) in HT29 (Figure 1A) and CCD 841 CoTr cells (Figure 2A) were not detected during the experiments.

A control time-lapse imaging experiment, using the same imaging parameters, did not show any harmful phototoxic effects for the studied cells. Manganese porphyrin fluorescence was not detected in either of the cell lines (Figure 1B and Figure 2B).

However, free-base porphyrin was quickly found in both cell lines just after the first excitation (2 h of incubation). Initially, it was localized at the edge of the cytoplasm and after further incubation in the perinuclear area C (Figure 1C and Figure 2C).

In UV–VIS absorption spectra of both normal and cancer cell culture media (Figure 3), distinct bands centered at 420–430 nm (Soret band), characteristic of porphyrins, were observed and a large signal was detected in the 250–300 nm spectral region. After 2 and 24 h incubation of colon cells with porphyrin, a decrease in the intensity of the Soret band was observed and it was more expressed in cancer cells than in normal ones. The 260–280 nm spectral region wavelength band was shifted from 262 nm to 274 nm after 2 h incubation of cancer cells with porphyrins.

### 2.2. Cell Migration Analysis

To examine cell migration, three parallel experiments were performed: a scratch assay, a circle assay (horizontal migration) and a vertical migration analysis. Horizontal migration is presented in Figure 4 (scratch assay) and Figure 5 (circle assay).

Both horizontal migration assays confirmed the HT29 cell line’s low mobility in the presence of porphyrins (Figure 6A,C and Figure 7A,C). The analyzed concentrations of porphyrins did not significantly influence the migration of cancer cells. In the case of the CCD 841 CoTr cells, both analyzed compounds significantly inhibited migration when combined with white light irradiation (Figure 6B,D and Figure 7B,D).

Vertical migration analysis demonstrated a significant increase in cell migration through a membrane with 8 µm pores compared to a membrane with 0.4 µm pores (Figure 8A,B). In contrast to the HT29 cells, CCD 841 CoTr cells are characterized by a higher mobility through a membrane with pores of 8µm after induction with light than non-illuminated cells, as compared to the control (Figure 8C,D).

In order to find possible changes in selected factors associated with cellular motility, two further tests were performed: F-actin filaments staining and signal transmission via connexons analysis. Porphyrins had no significant influence on the cytoskeleton filaments structure of the HT29 or CCD 841 CoTr cells, regardless of white light irradiation (Figure 9).

Lucifer Yellow staining (Figure 10A,B) demonstrated that signal transmission via connexons was limited to a short distance between cells (max: 40 µm for HT29 (A) and 30 µm for CCD 841 CoTr (B)). However, in cancer cells, the signal-transmission distance was shorter under white light irradiation, whereas in the case of the CCD 841 CoTr cells, the distance slightly increased after white light irradiation (Figure 11A,B). 

## 3. Discussion

In the present study, intracellular localization of the analyzed porphyrins and cell mobility under their influence was investigated. Evaluation was carried out using a laser scanning confocal microscope, among other tools. A porphyrin fluorescence signal was predominantly detected in the intracellular area and membrane localization could not be excluded. No manganese porphyrin fluorescence was found inside the cells. This phenomenon can be explained in several ways. Mn-porphyrin molecules did not show bright fluorescence emission at any excitation wavelength [22,23]. Coordinated metal ions increase the probability of non-radiative decay of the excited state of porphyrin [18]; therefore, fluorescence spectroscopy could not be employed for determining the localization of Mn-porphyrin molecules. However, cells absorbed the porphyrin [24]. Our previous study [25] indicated that the cytotoxic activity of manganese porphyrin was due to damage to the cell membrane. The evidence suggests that the porphyrin under investigation may not be entering the cell, but instead accumulating within the cell membrane or culture fluid. This could potentially lead to cell death via a non-photodynamic therapy mechanism from outside the cell. Alternatively, it is possible that the porphyrin is penetrating the cell, but not exhibiting fluorescence. The experiment also suggests that the Mn-porphyrin does not dissociate into its free-base form, i.e., it is stable in the colon cells and their environment. In this case, fluorescence from the demetallated porphyrin would likely be observed. The lack of fluorescence emission by porphyrin could also be caused by its interaction with biomolecules and other fluorescence quenching substances, e.g., proximity to other macrocyclic compounds [26].

The presence of hydrophilic groups in porphyrins makes them water-soluble. In turn, the presence of hydrophobic groups allows them to penetrate the cytoplasmic membrane and the inner membranes of the cell [27]. Hydrophilicity facilitates distribution and lipophilicity facilitates better cellular uptake of specific compounds [28]. For this reason, porphyrins first localize on the periphery of the cytoplasm and then, after further incubation, in intracellular compartments [29,30,31]. Kramer-Marek et al., using confocal laser scanning microscopy, determined the localization of lipophilic porphyrin derivatives on the periphery of the cytoplasm in the initial stage of the experiment and in the cytoplasm, near the cell nucleus, after further incubation [30]. A similar localization of cationic porphyrins has been demonstrated in other studies performed on cervical cancer cells. In the initial phase of the experiment, porphyrins were located on the periphery of the cytoplasm and, after further incubation, in intracellular compartments, such as the ER (around the nucleus), in mitochondria, or in the Golgi apparatus [29]. Cationic porphyrins (such as the FB used in this study) have a high affinity to plasma membranes [32]. It was observed that porphyrins interact with cancer cell membranes and cause their apoptosis. In the case of Tca8113 squamous cell carcinoma cells of the tongue, chlorine e6, after a longer incubation, was localized mainly in the mitochondria and less frequently in the lysosomes [33,34]. In turn, studies on electroporated colorectal cancer cells, LoVo 107 and LoVoDX, also confirmed the localization of manganese and cobalt porphyrins in the cytoplasmic space and in intracellular compartments, including mitochondria [31]. The results obtained in the presented research on the localization of free-base porphyrin in cells indicate that the primary place is the outer space of the cytoplasm and then the perinuclear space (see Figure 1 and Figure 2). Due to the presence of both peripheral hydrophilic groups and a hydrophobic core, free-base porphyrin can freely migrate through the cell membrane into the cell interior and then to other intracellular membrane structures. The uptake of porphyrins by colon cells, from the culture medium, was confirmed by porphyrin absorption spectra in the culture medium. A gradual decrease in the intensity of the porphyrin Soret band (420 nm), after 2 h incubation, indicates a decrease in its concentration. However, no significant decrease in porphyrin concentration was observed after 24 h of incubation; it remained at a similar (or a slightly higher) level as after 2 h of incubation. This may be related to the release of porphyrins from damaged cells or the cells reaching a possible maximum porphyrin accumulation value inside the cells within 2 h. It is worth noting that after 2 h of incubation of cancer cells, there was a greater decrease in porphyrin concentration in the culture medium compared to normal cells (see Figure 1 and Figure 2). Despite this, most of the porphyrin molecules are present in the culture medium and not in the cancer cells.

The next stage of the research was to determine the mobility of neoplastic and normal cells in the environment containing porphyrins. Studies have shown that light irradiation of cultures containing porphyrins significantly reduces the migration of normal cells, while cancer cells show little mobility compared to the control (control 24 h) (cell migration into the scratch and into the circle) (see Figure 5 and Figure 7). However, during vertical mobility analysis, the increased migration potential of the CCD 841 CoTr cells after white light irradiation was observed by their movement across the membrane (8 μm pores), which was in contrast to cancer cells. This phenomenon, i.e., the difference in cell migration in the scratch test and the circle migration test compared to the migration through the pores of the membrane after irradiation of cells containing porphyrins, may be related to a different form of cell migration in the above-mentioned methods (horizontal or vertical). The increased migration of neoplastic cells is closely connected with a significant risk of metastasis. In turn, inhibiting the migration of normal cells may be important, as it affects the tumor’s microenvironment, which stimulates its invasiveness. One of the factors which forms the tumor’s microenvironment and its malignancy is hyaluronic acid (HA). It has been shown that Mn porphyrins are able to catalyze the oxidative degradation of HA and thus reduce the invasiveness of the tumor [35]. In turn, MnTmPyP reduced the adhesion of MDA-MB-231 cells to collagen and fibronectin via sub-lethal H_2_O_2_ production. It is therefore indicated that inhibition of a hydrogen peroxide converting enzyme, such as manganese-dependent superoxide dismutase (MnSOD), and thus, an excess of H_2_O_2_, may block the migration of metastatic breast cancer cells [36]. In our research, we noticed that porphyrin affects the environment of the colon cells, probably by interacting with the components of the culture medium. This is evidenced by the shift of the wavelength band in the 260–280 nm spectral region (see Figure 3). It can be assumed that these interactions take place in the initial stage of porphyrin action on cells (up to 2 h). Other studies were performed by Wang et al., to determine mobility in an environment containing sinoporphyrin [37]. In a scratch and membrane transfer study (8 μm), it was shown that sinoporphyrin used in photodynamic therapy inhibits, depending on the dose, the migration of murine breast cancer cells (4T1) [37]. This porphyrin also inhibits the migration of breast cancer cells (MDA-MB-231) [38]. Similarly to our research, Reziwan Wufuer et al., also indicated that human colorectal cancer SW620 cell migration was inhibited by decreased pseudopodia numbers, under the influence of Ce6-PDT. The reasons for this, according to the authors, were microfilament depolymerization and the influence of porphyrin on the deregulation of the Rac1 expression [39]. Studies on the SW480 line of advanced colorectal cancer indicated that chlorine e6 inhibited tumor cell migration after white light induction [40]. In addition, it was shown that MnTE-2-PyP at a concentration of 30 µM inhibited in vitro human prostate cancer migration by the indicated mechanisms [41,42]. The mechanism of MnTE-2-PyP activity has also been shown to reduce tumor cell progression by reversing the EMT process. MnTE-2-PyP inhibited the Smad2/3 signaling pathway and the expression of matrix metalloproteinase 2 (MMP-2) and 9 (MMP-9) in colorectal cancer cells, reducing TGF-β-induced EMT and thus invasion of the colorectal and other cancer cells [43,44]. Moreover, manganese porphyrin reduced the chemotactic migration of neoplastic cells [45]. Similar observations were made by Chen et al., indicating a decrease in the MMP-2 and MMP-9 expression in osteosarcoma MG 63 cells subjected to a MPPa-PDT treatment [46]. Another study showed that manganese porphyrin (MnTE-2-PyP^5+^) significantly reduced the expression of mesenchymal markers but maintained an epithelial marker expression [47]. In turn, TMPyP4, a porphyrins derivative, is a quadruplex-specific ligand. It stabilizes G-quadruplex helices and inhibits the activity of the enzyme telomerase, which is overexpressed in >80% of cancer cells [21,48]. TMPyP4 also significantly reduced the integrin beta-1-mediated pathway [49]. Moreover, porphyrins can modify surface glycoproteins, as well as the structure and functionality of the lipid bilayer of the membrane and thus reduce the mobility of neoplastic cells [50]. However, studies conducted for this paper showed little mobility of the HT29 cells, both in the presence of porphyrins and without them. These findings align with previous research conducted on tumors, yet highlight a more progressed developmental stage. Moreover, they demonstrate that porphyrins employed in PDT can impede cell migration [37,38].

Actin is a retractile protein that builds filaments of myofibrils and the cell cytoskeleton’s microfilaments. Through polymerization, G-actin is converted into F-actin, forming the cytoskeleton [51]. The structure of the microfilaments, microtubules and intermediate filaments of the cytoskeleton is highly integrated and coordinated. These features are particularly important for maintaining, not only the shape, but also the viability of the cells. Any mutational changes or abnormal expression of cytoskeleton proteins may, on the one hand, affect apoptosis but, on the other hand, may be a significant factor influencing resistance to chemotherapy or the metastasis of cancer cells [52].

In order to test whether damage to the cell cytoskeleton by porphyrins could have an effect on the inhibition of cell migration, fluorescence staining of F-actin filaments was performed. No influence of the porphyrins at the concentrations studied (1 and 5 µM), on this structure, was demonstrated. However, literature data indicate that porphyrins can inhibit the migration of cells, e.g., by damaging the cytoskeleton [38,39,53]. An example is sinoporphyrin, which, in the case of a photodynamic reaction, inhibits the migration of human breast cancer cells (MDA-MB-231) by influencing the structure of F-actin filaments. DVDMS-PDT inhibited migration of the same cells in a manner closely related to the disruption of F-actin filaments [38]. Studies on the cellular actin cytoskeleton indicate that photodynamic therapy influences the reorganization of these filaments’ structure, reducing cell adhesion, while maintaining a constant level of this protein in the cell [52,54,55]. As a result of this activity, a reduction in cell proliferation and migration can be found. The changes observed by these authors were minor and were observed by the use of several dyes (FITC, DAPI) [56]. In addition, the active concentration of applied photosensitizer was significantly higher (15.2 μM) [56] than the porphyrins in this study (1 μM and 5 μM). Thus, the lack of action of the tested porphyrins on the actin cytoskeleton structure could be due to the low concentration of the compounds used.

In multicellular organisms, intercellular communication plays an important role. It regulates, among others, cell apoptosis, proliferation and differentiation. These interactions allow for rapid adaptation to changes in the cell’s microenvironment and for survival under stressful conditions. Interactions between cells in culture may also play an important role in inhibiting the migration of neoplastic cells. These interactions take place due to the presence of various adhesive molecules, receptors and connexon joints [57,58]. Connexons are transmembrane proteins that create connections between adjacent cells. These connections enable the transfer of an electrical signal, the flow of inorganic ions, secondary transmitters and small water-soluble particles. These channels enable intercellular communication [58]. In the present study, cells were stained with Lucifer Yellow, a dye with a molecular weight not exceeding 457Da, thus freely penetrating through the connexon channels. The obtained results indicate, however, that in the presence of porphyrins, connexon transmission does not play a significant role. The signal was transmitted only between closely adjacent cells. The relatively small distance of the transmitted intercellular signal may, however, be of significant importance, as demonstrated by the results of cell mobility in the scratch and the circle (see Figure 10 and Figure 11), in which, after exposure to the porphyrins, a strong inhibition of the migration of normal cells was observed.

## 4. Materials and Methods

### 4.1. Tested Substances and Experimental Conditions

5,10,15,20-Tetraphenyl-21H,23H-porphine manganese(III) chloride (further abbreviated as Mn) was purchased from Sigma-Aldrich Co. LLC, St. Louis, MO, USA at a purity of 95%, absorption in water with 1% methanol: λmax 467 nm, λmax 566 nm and λmax 600 nm.

5,10,15,20-Tetrakis(1-methyl-4-pyridinio)porphyrin tetra(p-toluenesulfonate) (further abbreviated as FB) was purchased from Sigma-Aldrich Co. LLC, St. Louis, MO, USA at a purity of 97%, absorption in water: λmax 421 nm and λmax 518 nm.

Predetermined amounts of the porphyrins were dissolved in dimethyl sulphoxide (DMSO, Sigma-Aldrich Co. LLC, St. Louis, MO, USA) to obtain stock solutions (100 mM—Mn porphyrin, 50 mM—FB porphyrin). Working solutions were prepared by dissolving stock solutions in the culture medium. A 0.1–10 μM range of porphyrin concentration was used in the experiments. However, depending on the type of experimental method, the concentrations used were appropriately selected, but still within the given range. During the incubation of cells with the porphyrins, culture plates were sealed with aluminum foil. Experiments were conducted in two separate conditions: no irradiation (during the entire experiment) and short exposure to white light (3 min) emitted by fluorescent lamps (Osram Lumilux Cool White, Wilmington, MA, USA) with a total power of 90 W and a photon flux density of 120 µmol/m^2^s, according to the method described by Buczek et al. [59] and Frant et al. [25].

A May–Grünwald stain, a Giemsa stain, a Phalloidin–Tetramethylrhodamine B isothiocyanate and Lucifer Yellow CH dipotassium salt were obtained from Sigma-Aldrich Co. LLC, St. Louis, MO, USA. Cell culture inserts and carrier plates (0.4 µm and 8 µm) were purchased from Thermo Fisher Scientific Inc., Waltham, MA, USA.

### 4.2. Cell Cultures

HT29 cell line (ATCC^®^ No. HTB—38™)—Homo sapiens, human colorectal adenocarcinoma (grade I, derived from a 44-year-old female adult) was cultured in a RPMI 1640 medium supplemented with a 10% fetal calf serum (FCS) (GibcoTM, Paisley, UK) and antibiotics (100 U/mL penicillin, 100 μg/mL streptomycin and 0.25 μg/mL amphotericin B) (GibcoTM, Paisley, UK) at 37 °C, in a humidified atmosphere with 5% CO_2_.

CCD 841 CoTr (ATCC^®^ No. CRL—1807™)—Homo sapiens, human normal colon epithelial cells (SV40 transformed, delivered from a 21 weeks of gestation female) were cultured in a RPMI 1640+DMEM (1:1) medium (Sigma-Aldrich Co. LLC, St. Louis, MO, USA) supplemented with 10% FCS and antibiotics at 34 °C, in a humidified atmosphere with 5% CO_2_.

### 4.3. Study of Porphyrin Uptake by Colon Cells

Analyses of subcellular localization of porphyrins in cancer and normal cells were conducted using laser scanning confocal microscopy. For this purpose, a LSM780 Zeiss microscopy system equipped with an AxioObserverZ.1 inverted microscope and a Plan-Apochromat 63×/1.40 Oil DIC M27 objective was used. This microscope allows the imaging of living cells for a long time, maintaining the desired constant temperature and CO_2_ concentration during the experiment.

For visualization of porphyrin accumulation, cells at density 5 × 10^4^ cells/mL, grown on glass bottom Petri dishes (35 mm; Greiner Bio-One GmbH, Kremsmünster, Oberösterreich, Austria), were incubated for 2 h with porphyrins (2% FCS; 1 μM and 10 μM Mn porphyrin; 1 μM FB porphyrin) in a culture medium, without phenol red, in dark conditions. The lack of this indicator in the culture medium is due to the fact that the excitation wavelength of 405 nm, used for viewing the porphyrin, is also in the range of the absorption band, in the UV–VIS absorption spectra of phenol red. Subsequently, the medium was discarded, replaced by the culture medium without porphyrins (also, a lack of phenol red), and cells were subjected to time-lapse live-cell imaging. The autofluorescence of normal colon epithelial cells and colon cancer cells was accounted for using cell suspensions that were not treated with a porphyrin. Cells not treated with porphyrins were also taken as a control with which the results obtained on analogous cells, after their incubation, with the tested macrocyclic compounds, were compared. Images of cells were collected every 10 min. The fluorescence of the porphyrins was collected with a PMT detector operating in a 600–700 nm range. A 405 nm diode laser was used for the excitation of FB porphyrin and a 458 nm laser light was used for the excitation of Mn porphyrin. The laser power for both of the porphyrins was the same and was set at 2%, in order to avoid photobleaching and phototoxic effects. Excitation wavelengths were from the range of the Soret band characteristic of the corresponding porphyrins. At the same time, images in the transmission light mode (DIC—differential interference contrast) were acquired. The experiments were repeated three times.

In order to confirm that porphyrins (1 µM) penetrate into colon cells, the presence of porphyrin in culture media was determined by absorbance spectroscopy. The UV-VIS absorption spectra of the culture media were recorded at the beginning of the culture and after 2 and 24 h of normal and cancer cell culture, respectively. The post-culture media samples were maintained in darkness, to prevent light activation until spectral measurements. A total of 1 mL of culture medium was placed in a 0.5 cm quartz cuvette and the absorption spectra of the porphyrins were measured/recorded against a reference sample (containing RPMI 1640 without phenol red 2% FCS and no porphyrins) with a JASCO V-730 spectrophotometer, in the 250–700 nm range.

### 4.4. Cell Migration into the Free Space (May-Grünwald-Giemsa Staining) (MGG Staining)

For the cell migration studies, the May–Grünwald–Giemsa staining was used. It also allowed for the visualization of morphological changes in the cells under the influence of the tested porphyrins. The use of these two dyes allows for the observation of the cell nucleus, which stains purple (May–Grünwald), and the cytoplasm, which is light pink (Giemsa) [60].

Staining was performed on plastic Petri dishes (35 mm) at room temperature. After incubating the cells with the analyzed substances, the medium was removed, and 1 mL of May–Grunwald dye was added, and incubation was carried out for 1.5 min at room temperature. An equal volume of distilled water was then added, and the liquid was removed from the plates after another 1.5 min of incubation. Cells were rinsed with distilled water (2–3 times) and stained for 20 min with a Giemsa stain (diluted 1:20 in distilled water). Following dye removal, the plates were washed three times with distilled water. The cells were analyzed under an Olympus BX51 microscope (Olympus, Tokyo, Japan).

#### 4.4.1. Scratch Assay

In order to evaluate the potential ability of the tested porphyrins to inhibit cell migration, a scratch assay was performed. It allows comparison between the number of migrating cells in the control group with the number of migrating cells in the tested one. The scratched area, made on a confluent cell monolayer, can be used to determine the rate of cell migration for a certain period of time [61]. In our study, the time of cell migration to the scratch was 24 h.

After receiving a confluent cell monolayer, a plastic pipette tip (P300) was used to scratch (725 μm scratch width), and porphyrins were added to the culture medium (2% FCS; 0.1 µM and 1 µM Mn porphyrin or 1 µM and 5 µM FB porphyrin). Simultaneously, the control of the scratch was dyed (MGG staining). The concentrations of the porphyrins were selected, based on their toxic activity in relation to the tested cells. The toxicity analyses have been described in a previous paper [25]. After 24 h of cell incubation, MGG staining was performed, and scratch diameter differences were measured. The analysis was performed with a light microscope, the Olympus BX51.

#### 4.4.2. Circle Assay

A modified scratch assay method, the circle assay (cell migration to the circle), was used to confirm the porphyrins’ potential ability to inhibit cell migration. In this method, the scratch was replaced by a circle. The assessment of the circle area, made on a confluent cell monolayer, allows determination of the rate of cell migration, after a specified period of time (24 h).

After receiving a confluent cell monolayer, the circle (4340 µm in diameter) was performed with a plastic pipette tip (P300) and next, porphyrins were added to the culture medium (2% FCS; 0.1 µM and 1 µM Mn porphyrin or 1 µM and 5 µM FB porphyrin). At the same time, a control circle was dyed (MGG staining). The concentrations of the porphyrins were selected based on their toxic activity, in relation to the tested cells. Toxicity analyses have been described in a previous paper [10]. After 24 h of cell incubation, MGG staining was performed and the differences in circle diameters were measured. The analysis was performed with a binocular Olympus SZX16.

### 4.5. Vertical Migration Analyses

The method based on inserts with porous membranes is commonly used to visualize vertical cell migration in laboratory studies. Cancer and normal cells are able to move; therefore, cell culture inserts are an ideal model of vertical cell migration [62]. In order to determine vertical cell migration, inserts with two sizes of pores were used: 0.4 µm and 8 µm. Smaller pores indicate more invasiveness of the cells (partial penetration), whereas larger pores allow determination of the number of cells that penetrated the membrane (cell mobility).

The cells (density 1.5 × 10^5^ cells/mL) were incubated for 24 h in inserts, in 24-well plates, in the culture medium. Next, the medium was removed and a 2% FCS culture medium with porphyrins (0.1 µM Mn porphyrin; 1 µM FB porphyrin) was added to the inserts, with a 4% FCS culture medium being added outside the chamber—into the well. After 24 h, the medium was removed, and the chambers were fixed with a 2.5% formaldehyde solution in PBS (with Mg^2+^ and Ca^2+^ ions) for 10 min. The cells were then dehydrated for 20 min, in 99.8% methanol. The next MGG staining was performed (20 min). At the end, cells which had adhered to the upper surface of the membrane were removed, using a cotton swab and discarded. Cells from outside the inserts were counted under a light microscope (Olympus BX51).

### 4.6. F-Actin Staining

In order to visualize the tested protein in the cell, a conjugated dye rhodamine-phalloidin was used. Phalloidin is an alkaloid derived from ‘Death Cap’ that binds to actin (strongly to F-actin), stabilizing its structure. Binding occurs at a 1:1 stoichiometric ratio, with one molecule of actin binding to one molecule of phalloidin. The cytoskeleton structure is then revealed by rhodamine as orange fluorescence (λexc = 570 nm) [63].

The cells (density 1 × 10^5^ cells/mL) were incubated for 24 h in 4-well Lab-Tek chamber slides (Nunc) filled with 1 mL of a culture medium. Next, the medium was removed and a 2% FCS culture medium with porphyrins was added (1 µM and 5 µM; Mn porphyrin and FB porphyrin). After 24 h of incubation, the cells were fixed with a 10% paraformaldehyde (*v*/*v*) solution in PBS (with Mg^2+^ and Ca^2+^ ions) for 20 min. Next, after fixative removal, the cells were rinsed 3 times in phosphate buffered saline (PBS), exposed to a Triton X-100 (0.2%, *v*/*v*) (Sigma) solution for 5 min and rinsed 3 times with PBS. Finally, the cells were incubated for 30 min with tetramethyl-rhodamine-isothiocyanatephalloidin (TRITC-phalloidin) (1 µg/mL) and incubated in the dark at 37 °C, in a humidified atmosphere of 5% CO_2_/95% air for 30 min. A fluorescent microscope (Olympus, BX51, Tokyo, Japan) (λexc = 570 nm) was used for examination of the cells. A qualitative analysis of the fluorescent images was conducted, using AnalySIS imaging software (Olympus, Tokyo, Japan).

### 4.7. Analyses of Gap Junction (Lucifer Yellow Staining)

This method is used to determine the existence of connections between cells and how far away from the cell an initial signal can be passed through a gap junction [57]. Most intercellular channels are filled with freely moving molecules with a mass of 1.000 Da. With a mass of 457 Da, Lucifer Yellow can easily penetrate the cells. If a cell absorbs the dye, it should transfer Lucifer Yellow to a neighboring cell (via a gap junction and diffusion). When excited by ultraviolet light, Lucifer Yellow emits a bright green fluorescence.

Porphyrins were added to the culture medium after obtaining a confluent cell monolayer (2% FCS; 0.1 µM and 1 µM Mn porphyrin; 1 µM and 5 µM FB porphyrin). After 24 h of incubation, the medium was removed, and the scratch was made using a sterile scalpel. Following the addition of Lucifer Yellow (0.5 mg/mL in PBS with Mg^2+^ and Ca^2+^), the cells were incubated for 3 min in the absence of light, then fixed with a 4% (*v*/*v*) paraformaldehyde solution in PBS (with Mg^2+^ and Ca^2+^ ions) and observed under an Olympus BX51 fluorescence microscope (UV light). Finally, the distance between the signal in the first cell (the nearest one to the scratch) and the signal in the last cell (farthest from the scratch) was calculated using the computer program, analySIS Image Processing imaging software.

### 4.8. Statistics

The results are presented as means ± SD of three independent experiments (n = 3). The data were analyzed using one-way analysis of variance ANOVA, followed by Bonferroni’s multiple comparison post hoc test in the GraphPad Prism 5.0 program. Differences of *p* ≤ 0.05 were considered significant.

## 5. Conclusions

In conclusion, these studies were successful in determining the localization of free-base porphyrin in the tested colon cells. Besides photodynamic effects, inhibition of cell migration by porphyrins can play a crucial role in the overall mechanism of colon tumor death. Due to the low concentrations of the applied porphyrins, no changes in the cellular cytoskeleton were observed. It has been shown that the mobility of normal cells exposed to porphyrins and white light irradiation is limited. Neoplastic cells, due to their origin from the early stages of tumor development, were not characterized by an intensive migration capacity; thus, no clear effect of porphyrins on this cell feature was observed. The presence of porphyrins at low concentration in the microenvironment of normal and neoplastic cells did not significantly affect transmission via connexons.

This work suggests that the influence of porphyrins on colon cell migration depends not only on the type of porphyrin (containing a metal ion or not) and the method of its excitation, but also on the type of cells they affect and, very importantly, the concentrations of these macrocyclic compounds.

Further studies based on different metal porphyrin complexes should be investigated to determine the differences between free-base and metallated porphyrins in the mobility of colorectal cancer cells both in the tissue and to distant organs.

## Figures and Tables

**Figure 1 molecules-28-02907-f001:**
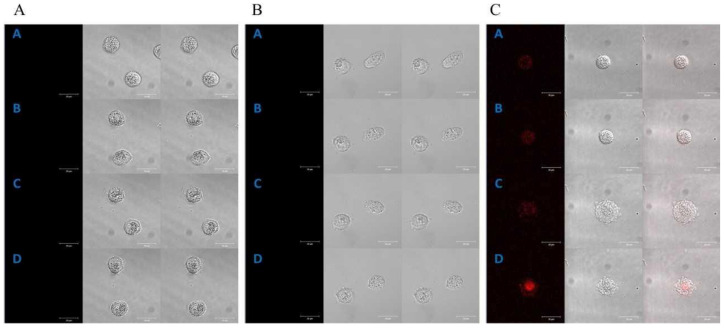
Control of the localization of porphyrins in the HT29 cell line (**A**); selected images received at the times: [A]—0 min, [B]—2 h 20 min, [C]—5 h and [D]—7 h 20 min. The localization of manganese porphyrin in the HT29 cell line (**B**); selected images received at the times: [A]—0 min, [B]—1 h 10 min, [C]—3 h 10 min and [D]—5 h 10 min. The localization of free-base porphyrin in the HT29 cell line (**C**); selected images received at the times: [A]—0 min, [B]—1 h 10 min, [C]—2 h 30 min and [D]—3 h 40 min; column 1: the fluorescence of porphyrin, column 2: the image in transmitted light, column 3: the image superimposed with columns 1 and 2. A concentration of 0.1 µM for both porphyrins was used in the study. Bar 20 μm.

**Figure 2 molecules-28-02907-f002:**
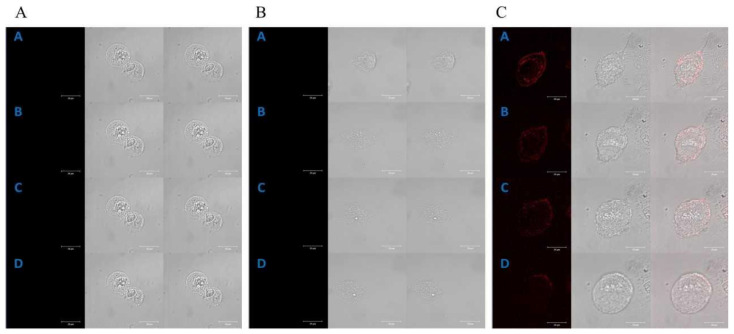
Control of the localization of porphyrins in the CCD 841 CoTr cells (**A**); selected images received at the times: [A]—0 min, [B]—2 h 20 min, [C]—4 h 50 min and [D]—6 h 40 min. The localization of manganese porphyrin in the CCD 841 CoTr cells (**B**); selected images received at the times: [A]—0 min, [B]—40 min, [C]—1 h 30 min and [D]—2 h 10 min. The localization of free-base porphyrin in the CCD 841 CoTr cells (**C**); selected images received at the times: [A]—0 min, [B]—20 min, [C]—50 min and [D]—1 h 30 min; column 1: the fluorescence of porphyrin, column 2: the image in transmitted light, column 3: the image superimposed with columns 1 and 2. A concentration of 0.1 µM for both porphyrins was used in the study. Bar 20 μm.

**Figure 3 molecules-28-02907-f003:**
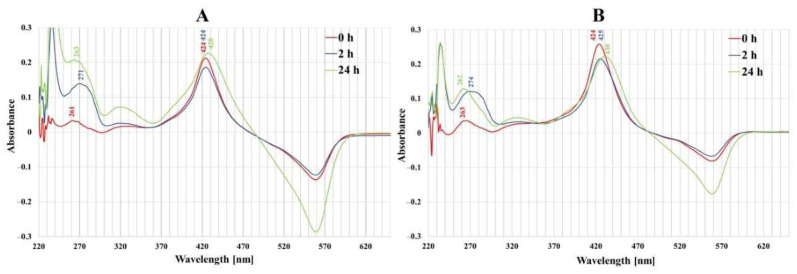
The UV-Vis absorption spectra of the free-base porphyrin (FB) in a post-culture media (RPMI 1640 without phenol red, 2% FCS) after 0 h, 2 h and 24 h of incubation of porphyrin with (**A**) normal colon epithelial cells (CCD 841 CoTr) and (**B**) colon cancer cells (HT29). Initial porphyrin concentration—1 µM.

**Figure 4 molecules-28-02907-f004:**
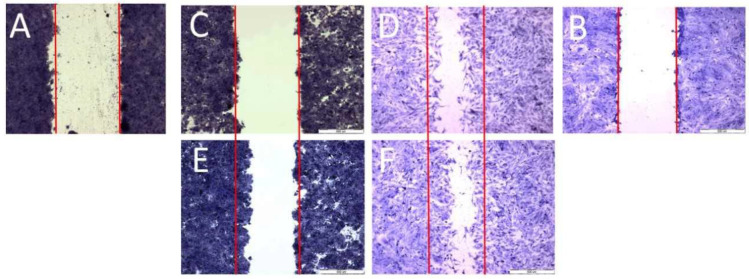
Scratch assay; scratch control: (**A**)—HT29, (**B**)—CCD 841 CoTr; control 24 h: (**C**)—HT29 and (**D**)—CCD 841 CoTr; scratch after 24 h: (**E**)—HT29 and (**F**)—CCD 841 CoTr. Bar = 500 μm.

**Figure 5 molecules-28-02907-f005:**
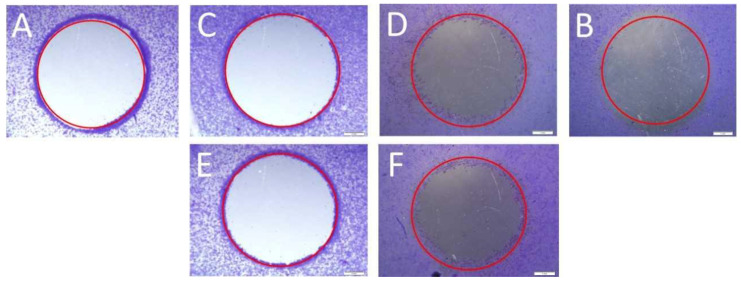
Circle assay; circle control: (**A**)—HT29, (**B**)—CCD 841 CoTr; control 24 h: (**C**)—HT29 and (**D**)—CCD 841 CoTr; circle after 24 h: (**E**)—HT29 and (**F**)—CCD 841 CoTr. Bar = 1 mm.

**Figure 6 molecules-28-02907-f006:**
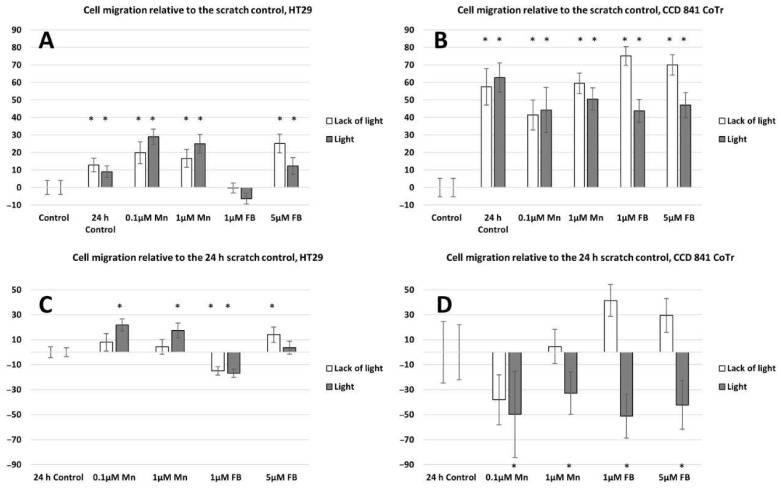
Percent of cell migration relative to the scratch control: (**A**)—HT29 and (**B**)—CCD 841 CoTr; cell migration relative to the 24 h scratch control: (**C**)—HT29 and (**D**)—841 CoTr. A negative percent means inhibiting the migration. Columns and bars show the mean ± standard deviation (n = 3). Statistical significance was analyzed at *p* < 0.05. * Statistical significance in comparison to appropriate control at *p* < 0.05.

**Figure 7 molecules-28-02907-f007:**
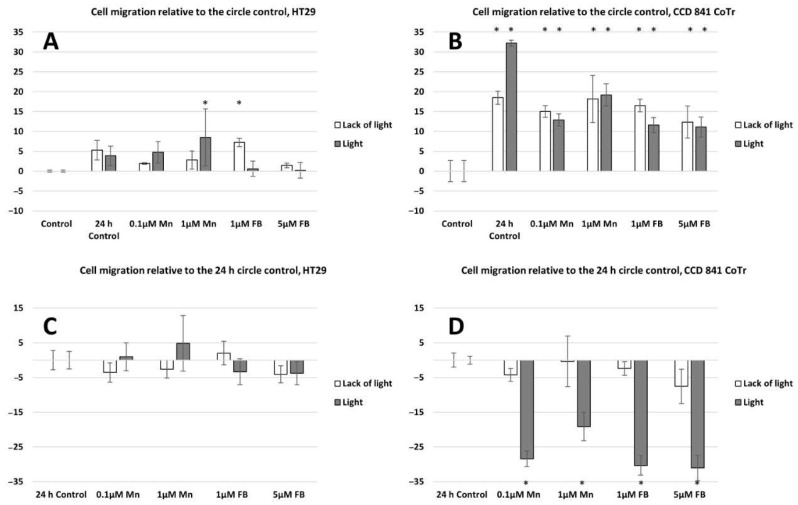
Percent of cell migration relative to the circle control: (**A**)—HT29 and (**B**)—CCD 841 CoTr; cell migration relative to the 24 h circle control: (**C**)—HT29 and (**D**)—841 CoTr. A negative percent means inhibiting the migration. Columns and bars show the mean ± standard deviation (n = 3). Statistical significance was analysed at *p* < 0.05. * Statistical significance in comparison to appropriate control at *p* < 0.05.

**Figure 8 molecules-28-02907-f008:**
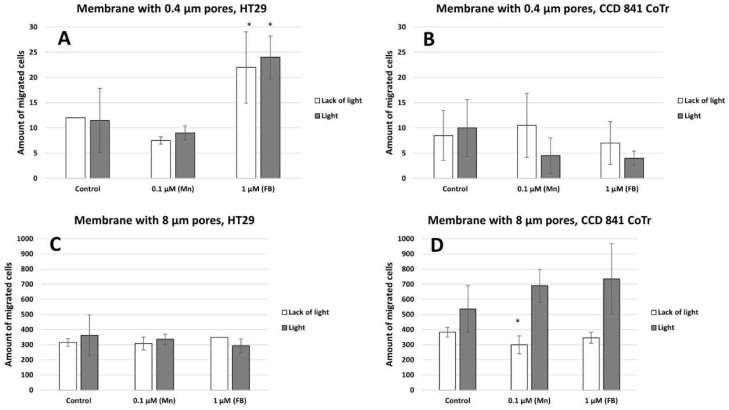
Amount of cells migrated to a membrane with 0.4 µm pores (**A**)—HT29 and (**B**)—CCD 841 CoTr, and a membrane with 8 µm pores (**C**)—HT29 and (**D**)—CCD 841 CoTr. Columns and bars show the mean ± standard deviation (n = 3). Statistical significance was analysed at *p* < 0.05. *Statistical significance in comparison to appropriate control at *p* < 0.05.

**Figure 9 molecules-28-02907-f009:**
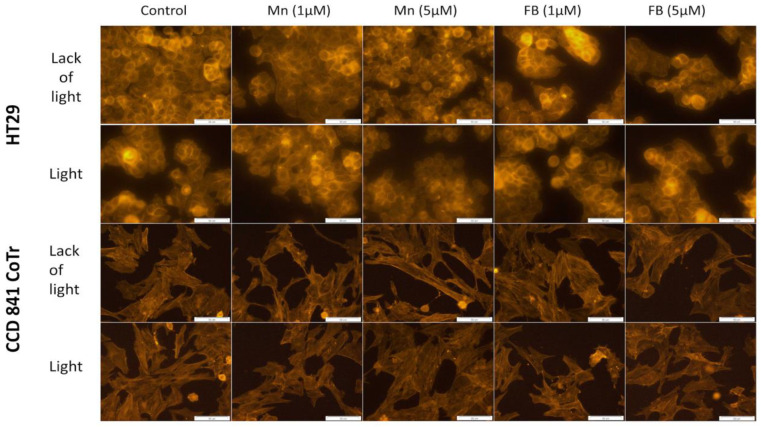
F-actin staining; first column: control of cells, second column: Mn porphyrin 1 µM, third column: Mn porphyrin 5 µM, fourth column: FB porphyrin 1 µM and fifth column FB porphyrin 5 µM; first line: no irradiation conditions (HT29), second line: white light irradiation (HT29), third line: no irradiation conditions (CCD 841 CoTr) and fourth line: white light irradiation (CCD 841 CoTr). Bar 50 µm.

**Figure 10 molecules-28-02907-f010:**
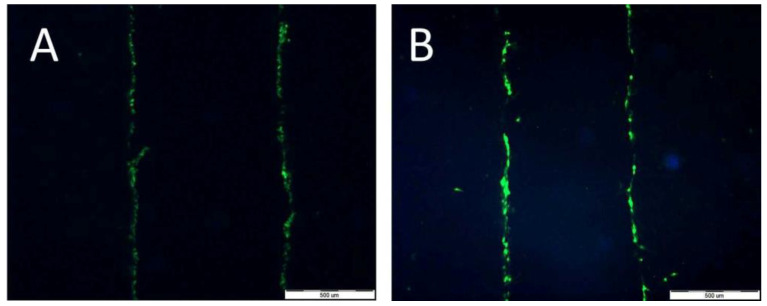
Signal transmission between HT29 cells (**A**) and CCD 841 CoTr cells (**B**). Bar 500 µm.

**Figure 11 molecules-28-02907-f011:**
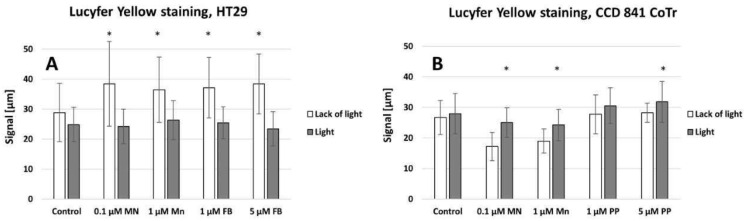
The signal transmitted between cells (**A**)—HT29 and (**B**)—CCD 841 CoTr. Columns and bars show the mean ± standard deviation (n = 3). Statistical significance was analysed at *p* < 0.05. * Statistical significance in comparison to appropriate control at *p* < 0.05.

## Data Availability

Data are available from the authors.

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
