# Peer review of "Cellular Localization of Selected Porphyrins and Their Effect on the In Vitro Motility of Human Colon Tumors and Normal Cells"

_molecules, 2023, doi:10.3390/molecules28072907_

Round 1

Reviewer 1 Report

Abstract:

1.       The abstract was not clear and the objective of the paper was not clearly validated from the abstract.

2.       The future perspective of the experiment should be mentioned in the abstract.

Introduction:

1.       The literature from past work done in the same field missing to strength the introduction section.

2.       There are many sentences that required proper citation. Do revise it.

3.        The authors should refer to more new literature in Introduction and keep abreast of the latest research trends related to the use of different types of compounds  as anticancer

[https://www.mdpi.com/1420-3049/24/13/2511; https://www.sciencedirect.com/science/article/abs/pii/S101060302100441X;

https://www.sciencedirect.com/science/article/abs/pii/S0022286022015952].

Results and Discussion:

1.           The result and discussion section needs to be elaborated more.

2.         The discussion did not provide specific reasons for the results. The provided explanation could have been more satisfactory also.

3.       The strong hypothesis, scientific facts, and validation of previous reports are entirely missing. You must rewrite it and cite the recommended papers listed in the comment section of this review report.

4.       The authors must tell the audience why chose the colon cell line not the other.

5.       The reference drug must be involved in this study

6.       Compare your material's efficiency with other published articles for the same target.

Conclusion:

1.       In the conclusion section the authors have only mentioned the data but major finding is missing from the conclusion part. Need to rewrite and incorporate this important concern of reviewer.

2.       The conclusion section seems like abstract so there is a need to completely rewrite the conclusion part.

Author Response

Reviewer 1. Replies to remarks and comments

We thank the Reviewer for time and comments that allowed us to improve our manuscript. The manuscript was re-checked by a native speaker.

  1. The abstract was not clear and the objective of the paper was not clearly validated from the abstract.

We have supplemented the abstract with the purpose of the study introducing it in the end of the paragraph. It is now as follows: The aim of the present study was to analyse intracellular localization of tested porphyrins and their influence on mobility of cells after irradiation with harmless white light.

  1. The future perspective of the experiment should be mentioned in the abstract.

Due to the limited number of words that could be used in the abstract, we have included the relevant information in the Introduction section. It is as follows: The conducted tests may be helpful in the diagnosis of colorectal cancer, as well as an alternative in limiting the spread of this cancer already at the screening stage.

  1. The literature from past work done in the same field missing to strength the introduction section.

We have added the sentence presented below:

Among cancers which are the most common in humans the third one is colorectal cancer. In 2020 there were about 0.9 million deaths for this cancer worldwide (with 1.9 million cases). The estimation indicate that number of cases of colorectal cancer might be even double in 2040 [A]. Despite the fact that the colorectal cancer is located inside the human body which makes difficult for delivery of the light, however there were trials with porphyrin based substances which gave promising results as a photosensitizers in PDT [B, C, D].

A: Xi, Y., Xu, P. Global colorectal cancer burden in 2020 and projections to 2040. Transl. Oncol. 2021, 14(10), 101174. doi: 10.1016/j.tranon.2021.101174.

B: Dandash, F., Leger, D.Y., Diab-Assaf, M., Sol, V., Liagre B. Porphyrin/Chlorin Derivatives as Promising Molecules for Therapy of Colorectal Cancer. Molecules. 2021, 26(23), 7268. doi: 10.3390/molecules26237268

C: Janas, K., Boniewska-Bernacka, E., Dyrda, G., SÅ‚ota, R. Porphyrin and phthalocyanine photosensitizers designed for targeted photodynamic therapy of colorectal cancer. Bioorg Med Chem. 2021, 30, 115926. doi: 10.1016/j.bmc.2020.115926

D: Bretin, L., Pinon, A., Bouramtane, S., Ouk, C., Richard, L., Perrin, M.L., Chaunavel, A., Carrion, C., Bregier, F., Sol, V., Chaleix, V., Leger, D.Y., Liagre, B. Photodynamic Therapy Activity of New Porphyrin-Xylan-Coated Silica Nanoparticles in Human Colorectal Cancer. Cancers (Basel). 2019, 11(10), 1474. doi: 10.3390/cancers11101474

  1. There are many sentences that required proper citation. Do revise it.

We have added the references which, in our opinion, were missing in the introduction.

Liu, Q.Q., Zeng, X.L., Guan, Y.L., Lu, J.X., Tu, K., Liu, F.Y. Verticillin A inhibits colon cancer cell migration and invasion by targeting c-Met. J Zhejiang Univ Sci B. 2020, 21(10), 779-795. doi: 10.1631/jzus.B2000190.

Lai, X., Li, Q., Wu, F., Lin, J., Chen, J., Zheng, H., Guo, L. Epithelial-Mesenchymal Transition and Metabolic Switching in Cancer: Lessons From Somatic Cell Reprogramming. Front. Cell Dev. Biol. 2020,  8, 760. doi: 10.3389/fcell.2020.00760

  1. The authors should refer to more new literature in Introduction and keep abreast of the latest research trends related to the use of different types of compounds  as anticancer

[https://www.mdpi.com/1420-3049/24/13/2511; https://www.sciencedirect.com/science/article/abs/pii/S101060302100441X;

https://www.sciencedirect.com/science/article/abs/pii/S0022286022015952].

We have added the suggested sentences in the Introduction section:

A: Flefel, E.M.; El-Sofany, W.I.; Al-Harbi, R.A.K.; El-Shahat, M. Development of a Novel Series of Anticancer and Antidiabetic: Spirothiazolidines Analogs. Molecules 2019, 24, 2511. doi: 10.3390/molecules24132511

B: Abdelhameed, R.M., Abu-Elghait, M., El-Shahat, M. Engineering titanium-organic framework decorated silver molybdate and silver vanadate as antimicrobial, anticancer agents, and photo-induced hydroxylation reactions. J. Photochem. Photobiol. A. 2022, 423, 113572. doi: 10.1016/j.jphotochem.2021.113572

C: El-Sofany, W.I., El-sayed, W.A., Abd-Rabou, A.A., El-Shahat, M. Synthesis of new imidazole-triazole-glycoside hybrids as anti-breast cancer candidates. J. Mol. Struct. 2022, 1270, 133942. doi: 10.1016/j.molstruc.2022.133942

  1. The result and discussion section needs to be elaborated more.

We have made changes to the results section. Figures 4 and 5 were supplemented with additional controls that appeared in the figures 6 and 7. The Introduction section has been supplemented with additional descriptions. However, Due to the fact that the discussion is quite an extensive chapter and the opinions of other reviewers indicating that we compared results rigorously with the literature in order to give a plausible explanation of the results, we tried to include all responses to the reviewer's comments in the introduction of the manuscript.

  1. The discussion did not provide specific reasons for the results. The provided explanation could have been more satisfactory also.

StaraliÅ›my siÄ™ wyjaÅ›nić otrzymane wyniki zarówno w kontekÅ›cie wÅ‚aÅ›ciwoÅ›ci fizykochemicznych i biologicznych porfiryn, jak i specyfiki badanych komórek. Dodatkowe wyjaÅ›nienia zawarto we wstÄ™pie pracy.

  1. The strong hypothesis, scientific facts, and validation of previous reports are entirely missing. You must rewrite it and cite the recommended papers listed in the comment section of this review report.

Zasugerowane prace zostały dołączone do manuskryptu.

  1. The authors must tell the audience why chose the colon cell line not the other.

In Introduction section we have added the following sentence:

Among cancers which are the most common in humans the third one is colorectal cancer. In 2020 there were about 0.9 million deaths for this cancer worldwide (with 1.9 million cases). The estimation indicate that number of cases of colorectal cancer might be even double in 2040 [Xi Y, et al., 2021].

Xi, Y., Xu, P. Global colorectal cancer burden in 2020 and projections to 2040. Transl. Oncol. 2021, 14(10), 101174. doi: 10.1016/j.tranon.2021.101174.

  1. The reference drug must be involved in this study

In our study we analysed the effect of two different types of porphyrins (free-base and metal) in different light conditions. For our research as a control we have used normal/regular colon cells. Our research was not intended to compare to the other drug but to the effect to normal cells and to the lack of light conditions. In addition, this manuscript is part of a larger research effort, the first results of which have been already published [Molecules 2022, 27, 2006. https://doi.org/10.3390/molecules27062006]. There were carried out experiments related to the cytotoxicity of the tested porphyrins under specific experimental conditions. In those studies, clinically used cytostatics were used for comparison. These were camptothecin (CPT-11), 5-fluorouracil (5-FU) and leucovorin. We did not want to duplicate research related to cytostatics because that was not the main point here. We felt that the additional results might confuse and hinder the reader's perception of these results. However, we agree with the reviewer that in new substances toxicity studies, specific analyses should be performed comparing them to known and already used substances. For this reason, in the first part of our project which was already published, this type of analysis was performed.

  1. Compare your material's efficiency with other published articles for the same target.

We discussed this issue among the co-authors and came to the conclusion that the discussion is already quite a large chapter of our work. However, in the introduction, we introduced new citations, which we believe exhaust the element of comparing our compounds efficacy with selected other substances.

  1. In the conclusion section the authors have only mentioned the data but major finding is missing from the conclusion part. Need to rewrite and incorporate this important concern of reviewer.

Conclusion section has already been corrected.

  1. The conclusion section seems like abstract so there is a need to completely rewrite the conclusion part.

Conclusion section has already been corrected.

Author Response

Reviewer 2. Replies to remarks and comments

We thank the reviewer for his time and comments that allowed us to improve our manuscript. The manuscript was re-checked by a native speaker.

  1. From the title of the paper, it seems that the Mn porphyrin used is the metallated counterpart of the free base porphyrin used in the current research, which is not actually the case, and the reader can only find this out later at the end in the section ‘Materials and methods’. The authors should include the full names of free base and Mn porphyrins in the last paragraph of the introduction, where they write the aim of the research.

We agree with the reviewer's comment that Mn porphyrin is not the strict counterpart of the free base porphyrin. Therefore, we have added the full names of both porphyrins studied (5,10,15,20-tetraphenyl-21H,23Hporphine manganese(III) chloride, and 5,10,15,20-tetrakis(1-methyl-4-pyridinio)porphyrin tetra(ptoluenesulfonate ) in the introduction. The use of these porphyrins in the current study was a consequence of the continuation of our previous research, which was published in Molecules. We decided to test porphyrins with a similar structure (porphyrin Mn and porphyrin FB) to compare the different effects and factors involved in cancer and normal cells death/apoptosis under the influence of metallated and free-base porphyrins. Additionally we wanted to observe whether some changes do not result from the intrinsic nature of porphyrins, and not only from phototoxic effects and different photochemical reactions. Therefore, we also conducted experiments with or without irradiation.

  1. On the same note, there is no explanation for why they used different free base and the metallated porhyrin. Since the two porphyrins differ not only by the presence or absence of the Mn center but also by their porphyrin skeleton, meaning there are two variables. So how the true effect of the free base or metallated porphyrin can be judged?

We tried to select porphyrins that are characterized by a common architecture, have the same type of structure (A4 meso-substituted porphyrins), have the lipophilic macrocyclic structure, differ in the presence of the metal ion, and at the same time show quite good solubility in polar solvents (water, DMSO, acetone, THF). We would like to note that we could not eliminate some variables, such as substituents in the porphyrin ring, axial ligands, and counterions, which, together with the metal ion, could also affect the electron donor-acceptor properties of porphyrins.

We could not use the free-base counterpart of Mn porphyrin in this study because non-metalated porphyrin bearing phenyl substituents (H2TPP) is strongly hydrophobic and unfortunately does not dissolve in hydrophilic solvents. We based our findings on commercially available porphyrins, which have already been used in different biomedical studies. Both porphyrins have been found to display anticancer properties; however, the effects of both porphyrins have not been investigated on cell migration in human colon tumor and normal cell cultures in the context of the effect of white light irradiation. In our opinion, when using irradiation of cells with light or not, despite differences in substituents in the porphyrin ring (phenyl or pyridyl groups), it can be assumed that some differences in the observed effects are due to the presence or absence of metal ions in the porphyrin structure.

  1. Finally, even though in the discussion part, the authors have compared their results rigorously with the literature in order to give a plausible explanation of their results, the reason why in particular, for the metallated version, Mn porphyrin is used and not any other metals is not mentioned in the introduction. Therefore, the authors should discuss why they chose Mn porphyrin for their study.

Manganese porphyrins have been widely investigated in many fields, including chemical catalysis, materials science, and medical treatments.

It has been reported that in HeLa cells (5 μM, 6 h), Mn(TPP)Cl caused cytoprotective autophagy and immunogenic cell death. Proteomic analysis indicated that [Mn(TPP)Cl] impaired/disrupted several physiological processes, including DNA synthesis, transcription, mitochondrial respiration, RNA translation, and immune response (Wang F-X., et al., CCS Chem. 2021, 3, 2527–2537).

In turn, 5,10,15,20-Tetrakis(1-methyl-4-pyridinio)porphyrin is reported to be a promising photodynamic therapy photosensitizer. Moreover it is one of the most studied G-quadruplex porphyrin ligands, used as a model to show that the ligands can exhibit different binding features with different conformations of a human telomeric specific sequence (Chen, J., et al., Cell Proliferation. 2021, 54, e13101).

However, the effects of both porphyrins have not been investigated on cell migration in human colon tumor and normal cell cultures in the context of the effect of white light irradiation.

  1. I believe in the heading “…of a human colon tumor and normal cells”, ‘a’ is a typo error as it

doesn’t seem grammatically correct if used for the subject ‘cells’.

It is already corrected. The letter ‘a’ Is removed.

  1. Fig 1 caption: Change hrs./hr. to h and mins. to min

It is corrected. All changes are made.

  1. Place the captions of Fig 1 and 2 after their relevant images.

Thank you for pointing out this shortcoming. It has already been corrected.

  1. Line 88: The language of the sentence is not adequate. So, change the sentence “The presence

of manganese porphyrin fluorescence in both cell lines (Fig. 1B and Fig. 2B) was not detected.”

to “Manganese porphyrin fluorescence was not detected in both cell lines (Fig. 1B and Fig. 2B).”

Thank you for the correction. The sentence in the new wording has been included in the text.

  1. Line 94: Correct the terminology “absorption UV-Vis spectra” to “UV-Vis absorption spectra”.

Indeed, our definition was not entirely precise. Relevant suggested corrections have been made throughout the text.

  1. Please assign the “short” wavelength bands.

This wavelength band is in the 260–280 nm spectral region. We decided that the word “short” could be removed.

  1. In Fig 4 and 5 captions, put A, B, C, D in brackets or consider any other presentation style.

Thank you for the hint. In the figure 4 and 5 caption, the letters A,B,C and D are placed in brackets. Moreover, these figures have been supplemented with additional control which could be found in figures 6 and 7.

  1. In captions of both Fig 6 and 7, put A, B, C, D in brackets or consider any other presentation

style.

Thank you for the another hint. In the figure 6 and 7 caption, the letters A,B,C and D are placed in brackets.

  1. Fig 9 caption: change the language of the phrase ‘lack of light conditions’ to ‘no irradiation’.

Change ‘light conditions’ to ‘xxx nm light irradiation’.

Also, change similar phrases used elsewhere in the paper.

Suggested corrections have been made in the text. We also checked the text again to eliminate any similar abbreviations or shortcomings.

  1. Line 185, 186: “……CoTr cells, the distance was insignificantly further after induction with light (Fig. 11A-B).” Change it to- “……CoTr cells, the distance slightly increased after light irradiation (Fig. 11A-B).”

Thank you for suggestion. Appropriate changes have been made to the text.

  1. Heading 4.1. Tested substances and general: the used term ‘general’ is vague. Rewrite the

heading for complete information.

We agree that the term 'general' may have been vague. Therefore, we changed the heading to the following, which we believe reflects the content of this section: 4.1.Tested substances and experimental conditions

  1. line 364: Rewrite the phrase “…further used in short Mn…” to “abbreviated as Mn” (moreover,

in the paper, you have written the metallated one as “Mn porphyrin” not just “Mn”). Change

similarly in line 367.

Suggested changes have been made to the text.

Reviewer 3 Report

In this manuscript, the authors determined the localization of free base porphyrin in the tested colon cells and demonstrating the effect of porphyrins on the inhibition of cell migration. However, some important issues should be further addressed. As followings I will list my concerns.

1.     In cellular localization experiments, there were differences in time points selection for imaging a type of cells and for a porphyrin in different type of cells. Should there be a standard for selection or additional time points?

2.     The authors should double-check the figure format in the manuscript. 

3.     Figures 4 and 5 show the results of the horizontal migration assays. Although the later bar graphs compare the results of the experimental and control groups after 24 hours, a set of images of the control group after 24 hours should be added here.

4.     In cellular localization experiments, the concentration of porphyrin selected for uptake imaging was not clearly labelled. Did the uptake process depend on the porphyrin concentration?

Author Response

Reviewer 3. Replies to remarks and comments

We thank the reviewer for his time and comments that allowed us to improve our manuscript. The manuscript was re-checked by a native speaker.

  1. In cellular localization experiments, there were differences in time points selection for imaging a type of cells and for a porphyrin in different type of cells. Should there be a standard for selection or additional time points?

The choice of different time points was determined by porphyrin and cell type. The free-base (FB) porphyrin after light induction killed the cells too quickly, therefore the selected times of incubation were shorter. The longer time for cells incubation with the manganese (Mn) porphyrin was selected to check whether it did not cause an observable, significant effect after a longer incubation time compared to the free-base porphyrin. With the normal CCD 841 CoTr cells, there was an additional problem of mobility. Cells in the presence of Mn porphyrin were very mobile and difficult to maintain them  in the field of view of the microscope. Thus, during the analysis, it was even technically necessary to use different times of incubation for both porphyrins.

  1. The authors should double-check the figure format in the manuscript.

Thank you for pointing out the format of the figures. We analysed them by adjusting the resolution to the requirements of the journal.

  1. Figures 4 and 5 show the results of the horizontal migration assays. Although the later bar graphs compare the results of the experimental and control groups after 24 hours, a set of images of the control group after 24 hours should be added here.

We agree with the Reviewer that the images in Figures 4 and 5 should be adjusted to the data presented in Figures 6 and 7. Therefore, we added appropriate image controls in Figures 4 and 5 and supplemented the figure caption with appropriate description.

  1. In cellular localization experiments, the concentration of porphyrin selected for uptake imaging was not clearly labelled. Did the uptake process depend on the porphyrin concentration?

The concentration of porphyrins for this experiment (cellular localization) was chosen as the lowest possible without causing a cytotoxic effect. For manganese (Mn) porphyrin it was finally 0.1 µM (higher concentration killed cells and prevented observation, for free porphyrin it was also concentration of 0.1 µM. For manganese (Mn) porphyrin we did not observe porphyrin uptake by cells, and its cytotoxicity was probably related to cell membrane damage. The free-base porphyrin showed cytotoxicity at the higher concentration without light induction. Therefore a concentration of 0.1 µM was chosen which killed the cell rapidly after light induction anyway. Appropriate Information regarding the applied concentrations of both porphyrins has been added to the description of Figures 1 and 2.

Round 2

Reviewer 1 Report

accept in present form 

Reviewer 2 Report

The clarifications for the concerns raised are satisfactory and the changes made are adequate. Hence I recommend the publication of this manuscript in Molecules.